# Predominance of Recombinant Norovirus Strains in Greece, 2016–2018

**DOI:** 10.3390/microorganisms11122885

**Published:** 2023-11-29

**Authors:** Nikolaos Siafakas, Cleo Anastassopoulou, Maria Lafazani, Genovefa Chronopoulou, Emmanouil Rizos, Spyridon Pournaras, Athanasios Tsakris

**Affiliations:** 1Clinical Microbiology Laboratory, ATTIKON University Hospital, 12462 Athens, Greece; maria_lafa@hotmail.com (M.L.); spournaras@med.uoa.gr (S.P.); 2Department of Microbiology, Medical School, National and Kapodistrian University of Athens, 11527 Athens, Greece; cleoa@med.uoa.gr (C.A.); atsakris@med.uoa.gr (A.T.); 3Biopathology Department, Athens Medical Center, 5-7 Distomou Str., 15125 Marousi, Greece; gchronopoulou@euroclinic.gr; 42nd Department of Psychiatry, ATTIKON University Hospital, 12462 Athens, Greece; erizos@med.uoa.gr

**Keywords:** norovirus, genetic recombination, epidemiology

## Abstract

GII.4 noroviruses have caused the overwhelming majority of norovirus-related gastroenteritis cases during the past two decades. However, a trend towards the emergence of new genotypes and novel GII.4 variants provided the impetus to explore further the changing patterns in norovirus epidemiology during the present study. Genotyping of 60 norovirus strains detected during a period of 33 months (January 2016–October 2018) was performed on the basis of the capsid VP1-coding ORF2 gene sequence. All norovirus strains detected were classified into seven genotypes, six of which belonged to genogroup GII. GII.2 was the dominant genotype till February 2017, whereas GII.4 prevailed thereafter. Most of the GII.4 strains were of the Sydney_2012 variant, whereas five strains could not be classified. Further recombination analysis at the ORF1/ORF2 gene junction revealed that 23 out of 24 strains were recombinant, thereby showcasing the significant role of genetic recombination in norovirus evolution and epidemiology. Continuous genomic surveillance and molecular characterization are essential for tracking norovirus evolution, which could contribute to the elucidation of new aspects of virus–host interactions that potentially affect host morbidity and epidemiology.

## 1. Introduction

Noroviruses are known as the most significant cause of outbreaks and sporadic cases of acute, non-bacterial gastroenteritis in both children and adults in healthcare and community settings [1]. They constitute a diverse group of viruses that belong to *Calciviridae*, a family of non-enveloped viruses. They have a single-stranded, positive-sense RNA genome that is approximately 7.5–7.7 kb long and contains three open reading frames (ORFs). ORF1 encodes a non-structural polyprotein that is cleaved into six non-structural proteins involved in viral replication, including the viral RNA-dependent RNA polymerase (RdRp). In contrast, ORF2 and ORF3 encode the structural proteins of the virus, the major capsid protein VP1 and the minor capsid protein VP2, respectively. Although little information is available about the function and significance of VP2 in norovirus biology, the capsid protein VP1 has been the subject of extensive research during the past years [2]. This capsid protein is further divided into an N-terminal shell (S) domain that surrounds viral RNA and a C-terminal protruding (P) domain consisting of P1 and P2 sub-domains [1]. The P2 sub-domain harbors an extensive degree of amino acid variability since it is the most surface-exposed portion of the capsid protein and, subsequently, subject to stronger evolutionary pressure from the host’s immune system.

Noroviruses infect a wide spectrum of mammalian hosts in different terrestrial, aquatic, and aerial habitats [3]. Their evolutionary success across such a diversity of ecological niches and hosts stems from the plasticity of their RNA genome, their resilient, non-enveloped structure, and their highly infective nature of transmission. Specifically, their rather short, positive-sense RNA genome dictates a significant degree of genetic variability, driven by mutation and recombination events that lead to the constant emergence of novel viral strains, highly adaptable to forces of selective pressure. Their robust, naked viral particles can survive in the environment for a long time, even several weeks, and they require a very low infectious dose for transmission; less than 10 copies with a short incubation period and prolonged shedding suffice for the continuation of the infectious cycle [1,4]. Finally, they are transmitted quite easily via the fecal–oral route, including via contaminated water sources, food, and contaminated surfaces [4].

Globally spread noroviruses constitute the most significant cause of outbreaks and sporadic cases of acute, non-bacterial, gastroenteritis in humans. Both children and adults may be affected in healthcare and community settings [5], despite the common excretion of noroviruses in the feces of asymptomatic individuals [1]. The disease is usually self-limiting, and patients usually recover within 2 to 3 days. Yet, noroviruses have been associated with significant outbreaks in nursing homes, hospitals, cruise ships, and military camps [1]. It is estimated that approximately 200,000 deaths of children less than 5 years of age per annum are attributed to norovirus infection in developing countries [6], whereas increased mortality in the elderly is reported in developed countries [7].

At least 10 genogroups (GI-GX) differing by about 40–60% in their amino acid sequence, and 49 genotypes with a 20–40% amino acid sequence difference on the basis of the VP1 capsid-coding region (ORF2) have been described so far [3]. This diversity in norovirus strains is generated by two mechanisms: point mutations throughout the ORF1 and ORF2 genes that encode the non-structural proteins and the major capsid protein (VP1), respectively, and genetic recombination events, most significantly across the ORF1/ORF2 gene junction [8]. The evolutionary rate for the VP2-coding ORF3 gene is even greater than that of the VP1-coding ORF2 and, therefore, it may also play a significant role in norovirus diversity and evolution that remains to be explored [2]. Despite the diversity of noroviruses, infections in humans are almost always caused by members of genogroups GII and GI, with members of genogroup IV being responsible for only a few outbreaks [9]. 

Even though these two mechanisms of genetic variability are universal for all noroviruses, they are not exploited by all genotypes in the same way. Both antigenic drift and antigenic shift in the protruding, hypervariable P2 region of the capsid protein of GII.4 strains lead to the rapid emergence of novel, immune escape pandemic variants in a cyclic fashion every 2 to 4 years [10]. As a result, GII.4 strains have caused the overwhelming majority of norovirus-related gastroenteritis sporadic cases and outbreaks worldwide during the past two decades [11]. In contrast, different variants of the other norovirus genotypes circulate simultaneously with limited changes in their VP1 [8]. However, the prevalence of pandemic GII.4 variants during the last decade has been challenged by a trend towards the domination of new genotypes, such as GII.17 and a recombinant variant of GII.2. For instance, a novel variant of GII.17 and a recombinant variant of GII.2 (GII.2_GII.P16) had an increased epidemic prevalence that replaced, to an extent, the current pandemic Sydney_2012 GII.4 variants approximately 7–9 years ago [12,13]. GII.4 strains that could not be classified as an already known pandemic variant also started to emerge 4 to 5 years after the emergence and prevalence of the Sydney_2012 variant [12].

Continuous genomic surveillance and comparative analysis of emerging pandemic noroviruses and strains that circulate at an endemic level constitute a prerequisite for a better understanding of norovirus evolution and epidemiology. Detection of genetic variation in the ORF1 and ORF2 genomic regions and comprehension of the mode in which it shapes virus–host interactions are central to this quest. Following a previous report on the molecular epidemiology of noroviruses in children with acute gastroenteritis in Greece [14], the present study attempted to explore further the more recent pre-COVID-19 pandemic changing patterns in norovirus epidemiology and evolution in the specific geographic area at the crossroads of Europe, Asia, and Africa.

## 2. Materials and Methods

### 2.1. Stool Specimens and Viral RNA Extraction

Sixty stool samples collected in sterile containers from 58 children aged 1 month to 14 years and from 2 adults aged 30 and 40 years, respectively, all with symptoms of acute gastroenteritis, were studied in retrospect. These samples initially tested positive for the presence of norovirus antigen during a period of 34 months (January 2016–October 2018), using a rapid immunochromatographic method (RIDA QUICK Norovirus, R-Biopharm AG, Darmstadt, Germany). 

Sample collection for laboratory diagnosis of norovirus infection formed part of the standard patient management, and thus, only oral informed patient consent was required. Access to patients’ data was restricted for employees directly involved in diagnosis and reporting. Patients’ data were anonymized before any analysis was conducted. Clinical sample collection did not involve any invasive procedures since the specimens constituted human excreta. According to the national law, no approval by the Ethics Committee for the retrospective analysis of anonymized data from human excreta is required.

The samples were stored at −80 °C until further processing for RNA extraction, as previously described [14]. In brief, approximately 2 g of each stool sample were homogenized by vigorous vortexing for 10 min in 15 mL test tubes containing 5 mL of PBS (Thermo Fisher Scientific Inc., Logan, UT, USA), 0.5 mL of chloroform (Merck KGaA, Darmstadt, Germany) and 2 g of glass beads (Witeg, Wertheim, Germany). Stool debris was removed by centrifugation at 1500× *g* for 30 min at +4 °C, and the supernatant was then carefully removed and used immediately for viral RNA extraction with the QIAamp Viral RNA mini kit (Qiagen, Hilden, Germany), according to the manufacturer’s instructions. All extracted RNA samples were then used either immediately or stored at −20 °C before further processing.

### 2.2. Genotypic Characterization of Norovirus Strainsand Phylogenetic Analysis

The genogroup and genotype of each norovirus strain were determined on the basis of the partial ORF2 gene sequence, which codes for the VP1 capsid protein. In more detail, two separate reverse transcription polymerase chain reaction (RT-PCR) protocols were applied to the extracted RNA samples, specific for either genogroup GI or GII noroviruses, respectively, as previously described [15,16], with minor modifications. In brief, the primer pair G1SKF/G1SKR was applied for the detection of GI noroviruses, yielding a 330 bp long PCR product, whereas a 344 nt long genomic region of GII noroviruses was amplified with the primer pair G2SKF/G2SKR. Figure 1 shows schematically the primers used for RT-PCR and genotyping and their relative position on the norovirus genome. The reverse transcription reaction that converted the extracted RNA into cDNA was performed with the aid of the SuperScript™ II Reverse Transcriptase (Thermo Fisher Scientific Inc., Logan, UT, USA). First, 20 pmol of the reverse primer G2SKR for GII noroviruses, or G1SKR for GI noroviruses, and 2 μL of 10 mM deoxynucleotides (dNTPs) were added to 10 μL of extracted DNA and incubated at 65 °C for 10 min, after which they were immediately placed on ice. A reverse transcription reaction mix was then produced containing RT Reaction Buffer 5x, 0.01 M DTT, 20 units of ribonuclease inhibitor (Promega Corporation, Madison, WI, USA) and 100 units of Moloney Murine Leukemia Virus reverse transcriptase (M-MuLV RTase) and added to the RNA, reverse primer, and dNTPs from the previous step, up to a final reaction volume of 20 μL. The reaction was performed at 42 °C for 50 min, and the M-MuLVRTase was inactivated by heating at 70 °C for 10 min. 

The produced cDNA was amplified by PCR using a reaction mixture of 50 μL/tube containing 5 μL 10× PCR buffer, 4 μL dNTPs 0,25 mM each, 2 μL MgCl_2_ 25 mM (yielding a final [MgCl_2_] = 2 mM), 25 μL RNase-free water, 2.5 units Taq Polymerase (HotStar Taq DNA Polymerase, Qiagen, Hilden, Germany), 50 pmol of each of the two primers and 10 μL cDNA. An initial step of heating at 95 °C for Taq polymerase activation preceded 40 cycles of denaturation (95 °C, 30 s), annealing (50 °C, 30 s), and extension (72 °C, 60 s), followed by a 15-min incubation at 72 °C to complete the extension of the primers. 

The reaction products were visualized by electrophoresis on 1% low-melting agarose gel (Metaphor FMC Bioproducts, Rockland, ME, USA) stained with 1 μg/mL of ethidium bromide. The RT-PCR products were excised from the electrophoresis gel with a clean scalpel and purified with the aid of a QIAquick gel extraction kit (Qiagen, Hilden, Germany). Both strands were sequenced by VBC-Biotech GmbH (Vienna, Austria), using the same set of primers implemented in the genogroup-specific RT-PCR. 

Genotypic characterization of the strains on the basis of their partial ORF2 nucleotide sequences was then performed using the Norovirus Genotyping Tool version 2.0 [17], provided by the National Institute for Public Health and the Environment in Holland (RIVM). Identification of the closest related norovirus strains that have been described to circulate globally and show the highest sequence similarity with the strains identified in the current study was performed by analysis with the BLAST alignment software, version 2.4.0 [18]. Comparative phylogenetic analysis of the different genotypes, which included both norovirus strains detected in this study and closely associated strains identified by BLAST analysis, was conducted by MEGA software, version XI [19]. The phylogenetic tree was constructed using the Neighbor-Joining algorithm, and its statistical significance was estimated by calculation of the confidence values for the groupings (bootstrap values) with 1000 pseudo-replicates. Further information about the GenBank Accession Nos of the strains used is provided in the Appendix A.

Estimation of time-scaled phylogeny for norovirus strains of the two most prominent genotypes on the basis of the partial ORF2 genomic sequences included many homologous sequences of norovirus strains of the same genotype which have been detected during the past 50 years. All sequences were trimmed for equal length with the aid of Clustal Omega multiple sequence alignment software (https://www.ebi.ac.uk/Tools/msa/clustalo/ (accessed on 3 September 2023)) [20], and the evolutionary timeline was estimated using the Bayesian Markov Chain Monte Carlo method implemented in the BEAST software, version 2.2.1 [21] and visualized in FigTree v1.4.3. Information about the isolates that were used, the year of their detection, and their GenBank Accession Nos for GII.4 and GII.2 norovirus strains, respectively, is provided in the Appendix A. The temporal signal evaluation for each MCC phylogenetic tree using the root-to-tip analysis implemented in the TempEst software, version 1.5.3, is also provided, as well as information about the parameters used for MCC tree construction, model selection, and temporal tree construction using the Bayesian method.

### 2.3. Detection and Analysis of Recombinant Strains

A second RT-PCR protocol was implemented in order to search for genetic recombination events. The anti-sense primer G2SKR was used as before for the genotyping RT-PCR, but now another primer, JV12, was used as previously described [22]. This set of primers is specific for the amplification of an approximately 1112 bp long genomic region that spans the ORF1(RdRp)/ORF2(VP1) gene junction (described in Figure 1). Reverse transcription and PCR conditions were the same as those of the genotyping RT-PCR, with the exception that the primer annealing temperature was 50 °C. Purification of PCR products and sequencing were carried out as before. Recombinant strains were first determined with the aid of the Norovirus Genotyping Tool [17], followed by further recombinant sequence analysis using the SimPlot software, version 3.5.1 [23].

## 3. Results

### 3.1. Norovirus Genotype Distribution

All norovirus strains detected were classified into seven genotypes, six of which belonged to genogroup GII, which was, thus, the most prevalent (58/60, 96.7%). GII.4 was the predominant genotype (33/60, 55.0%), followed by GII.2 (15/60, 25.0%). Other genotypes included GII.6 (5/60), GII.3 (2/60), and GII.7 (2/60), while one strain was identified as GII.14 (1/60). Two GI strains were genotyped as GI.1 (Figure 2a).

The chronological pattern of norovirus genotype detection throughout the study period is shown in Figure 2b. There were two major waves of norovirus detection, the first from August 2016 to January 2017 and the second from February 2018 to September 2018. Hence, no seasonal pattern of norovirus circulation was observed. Interestingly, an abrupt end to GII.2 strain detection towards the beginning of 2017 was recorded, and the genotype was not detected thereafter. In contrast, the incidence of GII.4 strains markedly increased during the 2nd norovirus wave.

Table 1 shows all the norovirus genotypes of the strains that were detected with respect to both the partial ORF1 (RdRp) and ORF2 (VP1) nucleotide sequences. GenBank Accession Nos and information about genetic recombination events and their breakpoints on the norovirus genome are also included. 

### 3.2. Epidemiologic and Phylogenetic Analysis

The partial ORF2 sequences of all isolates were highly similar (>95%) to the respective sequences of norovirus strains that circulated worldwide in outbreaks and sporadic cases of gastroenteritis during the same time period, either in outbreaks and sporadic cases of gastroenteritis, or in the environment. Figure 3 shows the phylogenetic relationships of all strains identified during the present study, in addition to reference norovirus strains of the same genotype that circulated in Greece in the past and additional reference strains that circulated globally on the basis of their partial ORF2 gene sequences. The genetic clusters that were formed in the phylogenetic tree were accurately specific for each genotype and reliably supported by high bootstrap values. Figure 4 shows the temporal history of each of the two most prominent genotypes identified during the present study on the basis of partial ORF2 gene sequences. Newly identified strains are included, along with norovirus strains previously detected in Greece and numerous other norovirus strains of the same genotype that have been circulating globally during the last 50 years. A pandemic variant-specific type of genetic clustering of GII.4 strains throughout the years was observed (Figure 4a). GII.2 strains seem to have diverged into two separate clusters around 2002 (Figure 4b). The first cluster diminished in approximately 2015, whereas the second cluster has evolved into several sub-clusters and has produced strains that continue to circulate till the present study. It was interesting to observe that both separate clusters included recombinant GII.2[P16] strains and that the first GII.2[P16] strain was detected in 2008, before its epidemic emergence.

Almost all GII.4 strains were sub-classified into the pandemic “Sydney_2012” variant, whereas five strains could not be classified to a specific pandemic GII.4 variant. For four of these strains, it was possibly due to the short length of obtained ORF2 gene nucleotide sequences that were not sufficient for accurate variant classification. Nevertheless, three of these strains were highly similar to other “Sydney_2012” norovirus strains that circulated worldwide. Intriguingly, the fourth strain was similar to a “Hunter_2004” pandemic variant that circulated more than a decade ago, although the short length of the available sequence of the strain did not allow for accurate sub-classification. On the other hand, despite the much larger and inarguably adequate for more detailed genotypic characterization, partial ORF1/ORF2 sequence, isolate “346504/ATH/GII.4/2018” could not be classified as any known GII.4 pandemic variant.

### 3.3. Recombination Analysis

A sufficiently long sequence that spanned the ORF1/ORF2 gene junction that would enable genetic recombination analysis was obtained for 24 strains. Figure 5 shows representative results of the SimPlot analysis performed for each of the recombinant types identified during the present study. The recombinant strains and their sequence exchange breakpoints on the genome are listed in Table 1. Five out of six GII.2 strains were recombinant of the GII.P16 ORF2 subtype, and all three GII.6 strains were identified as GII.P7. All 15 GII.4 strains analyzed were recombinant, with nine identified as GII.P16, five as GII.P31, and one as GII.P21. 

## 4. Discussion

Regarding norovirus circulation, two distinct patterns were observed during the study period of 33 months, as shown in Figure 2. The first wave of norovirus circulation lasted for 5 months, from August 2016 to January 2017, whereas the second had an 8-month duration between February and September 2018. Hence, norovirus detection rates did not show any seasonal pattern throughout the current study, as was also reported in a previous study about the epidemiology of noroviruses in children in South Greece between 2013 and 2015 [14]. Very little, or no virus detection at all, was reported before or after each of these two time periods of norovirus circulation, something which would suggest that the increased rates of norovirus detection during each of these two periods may be attributed to epidemic activity irrelevant to season. In contrast, previous studies have reported a seasonal pattern of norovirus infection that culminates from winter till early spring, with a decline in summer months [24,25,26]. It has been suggested that, due to the frequent waterborne nature of norovirus transmission, humid weather conditions and rainfall may be positively correlated with increased rates of norovirus transmission, and low temperatures also lead to people crowding and higher chances for human-to-human transmission [27]. However, contrary to these findings, other studies have reported on summer peak of norovirus infection due to a possible increased risk of water and food contamination in hot weather or no seasonal pattern of infection at all, as observed in the current study [25,26]. Perhaps isolated epidemic activity varies not only between different geographic areas, where prevailing climatic conditions are different, but more importantly, different genotypes, variants, and strains may possess correspondingly different traits of transmissibility, infectivity, and survival in the environment, something that warrants further detailed investigation.

Two issues of major importance regarding norovirus epidemiology were identified during the present study. The first one was the abrupt change in norovirus genotype predominance: GII.2 noroviruses prevailed from September 2016 to January 2017, whereas GII.4 strains dominated the last part of the study period, from February 2018 to September 2018. The second key finding was the abundance of norovirus recombinant strains. Of the 24 strains that were analyzed for recombination events, 23 were indeed found to be recombinant. This was not the first time that recombinant norovirus strains were detected and characterized in Greece. For instance, the circulation of a recombinant GII.6[P9] strain was detected in both sewage and clinical samples [28]. More recently, GII.13[P16] recombinant strains were identified during a waterborne outbreak in Northern Greece in 2015 [29]. However, the present study recorded more extensive data on both the changing patterns of norovirus epidemiology and the prevalence of genetic recombination amongst different genotypes, thereby showcasing the significant role of genetic recombination in norovirus evolution, diversification, disease, and epidemiology.

GII.4 viruses are known to accumulate mutations that lead to the periodic replacement of circulating antigenic variants and, consequently, also to shifts in the prevailing respective genotypes. In contrast, other genotypes exhibit a limited repertoire of genetic variants that may persist for many years with very low variation in the VP1-coding ORF2 gene [30]. Intriguingly, recombination events in the non-structural part of the norovirus genome of such strains played a significant role in the emergence and prevalence of other genotypes over pandemic GII.4 strains. The abrupt predominance of GII.17 strains in various Asian countries between 2013 and 2015, associated with different RdRp types, provides an important example [31,32]. The same happened with the emergence of epidemic GII.2 strains in late 2016, an occurrence that was associated with recombination events in the supposedly well-conserved RdRp; the newly derived strains were designated as GII.2[P16] [33]. A similar phenomenon was observed during the present study since recombinant GII.2[P16] strains were detected for the first time in September 2016, prior to their reporting on a global scale in late 2016. As previously stated and shown in Figure 4b, the first report about the identification of a GII.2[P16] strain was many years before its epidemic prevalence. Although these recombinant noroviruses did not completely replace GII.4 strains during this study, they at least transiently prevailed over the pandemic genotype for approximately 5 months in 2016, and then, their incidence suddenly diminished during the following year, as also reported by other worldwide studies [34]. However, unlike the recently emerged GII.17[P17] strains that predominated only in parts of Asia in the 2014–2015 season, the GII.2[P16] strains were almost simultaneously reported in Asia and Europe, indicating the fast spread of this genotype across continents [35]. But this was not the first time that GII.2[P16] strains were associated with outbreaks, albeit at a local geographic level, since epidemic spread of this genotype was reported for the first time in Osaka, Japan, during 2009–10 [36]. Figure 4b shows the detection of GII.2[P16] strains as early as 2008, but a divergence of two genetic clusters of GII.2[P16] before the first reports of outbreaks is also obvious. The first epidemic strains detected in Japan, as well as the globally spread GII.2[P16] epidemic strains of 2016–2017 and currently circulating strains, belong to a somewhat more heterogeneous genetic cluster of noroviruses (cluster II in Figure 4b). This is perhaps a mere example of how increased genetic variation promotes evolutionary fitness that enhances adaptation and ultimate survival, whereas the genetically less diverse strains of the other GII.2[P16] cluster gradually diminished due to a corresponding lack of evolutionary adaptability.

The emergence of novel, pandemic GII.4 variants that prevail in a cyclic manner may not be attributed only to antigenic variation in the P2 protruding region of the VP1 capsid protein; recombination in the ORF1 has also been proposed as an important factor in new GII.4 variant prevalence since several new GII.4 variants harbored a novel ORF1 gene [37]. Nevertheless, GII.4 of a certain pandemic variant but with different RdRps, may still be detected, as shown in the present study. GII.4 strains of the Sydney_2012 pandemic variant generally harbored three different ORF1 genes (P16, P31, and P21, respectively). GII.4[P16] was the most prevalent and replaced GII.2[P16] during the later period of the study. This novel norovirus recombinant strain, GII.4[P16], emerged in the United States in 2015 [38] and has been detected since then in both sporadic cases and outbreaks of acute gastroenteritis, as well as environmental water worldwide [34,39,40]. It has also been reported that GII.2[P16] emergence predated GII.4_Sydney_2012[P16], providing a hint about the possible role of these recombinant GII.2 strains in the emergence of the latter [41]. 

Almost all GII.4 strains detected in the present study belonged to the Sydney_2012 variant. It is astonishing that strains belonging to the Sydney_2012 variant are still prevalent and continue to be detected despite the periodic shift of GII.4 predominant variants, which generally occurs every 2–3 years (as in the cases of New_Orleans_2009, DenHaag_2006, or Hunter_2004). This pattern is reminiscent of the circulation of strains of the Grimsby_1996 variant in the past [30]. Nevertheless, novel GII.4 variants that cannot be classified into a known clade have already emerged a few years ago [14]. Classification to a novel variant, such as a GII.4 “Hong Kong” that circulated in Asia between 2017 and 2019, has been proposed by researchers in the field for unclassified GII.4 strains [42]. We need continuous surveillance and analysis of many more strains in order to decipher whether the prevalence of a new variant is imminent or has already started. Perhaps such unclassified strains constitute a prelude towards the circulation of new variants, something which is compatible with the suggestion that GII.4 variants diverge from one another long before their pandemic emergence. There may be undetected circulation of these pre-pandemic variants at a low level during the period between divergence and pandemic emergence, something denoted by the long branches in the relevant timeline phylogenetic trees [39], as was the case in the present study (Figure 4a). Perhaps this is the case not only for GII.4 pandemic viruses but for other genotypes as well, considering the appearance of the GII.2[P16] 8 years before its prevalence in late 2016.

So far, the importance of both host–virus interactions and genetic variation in two regions of the genome (ORF1 and ORF2) has been highlighted [8]. However, little information is available regarding the role of other than the RdRp-coding, non-structural genes on norovirus evolutionary success and epidemiology. Moreover, the incorporation of analyses involving the highly variable domains of the much-neglected VP2 capsid protein should also be very useful since VP2 appears to play a significant role in norovirus biology and evolution by interacting with VP1, contributing to viral replication functions and preventing antigen presentation during the immune response [2]. Perhaps it is not only genetic recombination events that shape norovirus evolutionary success since single-point mutations in the RdRp have been shown to affect transmissibility and replication efficiency, thus increasing evolutionary fitness and epidemic potential [43,44]. Therefore, we still have a lot to learn about the precise mechanism that leads to the emergence and prevalence of epidemic noroviruses and how their genetic variability is related to increased virulence and transmissibility in order to design and implement appropriate strategies for the control of norovirus infection.

## 5. Conclusions

The present study provided new information about the prevalent norovirus genotypes that circulated in Greece during the period before the COVID-19 pandemic. Moreover, the significance of genetic recombination events in the emergence of novel epidemic and pandemic strains was highlighted. Continuous surveillance and more complete genomic characterization of many norovirus strains are essential for tracking norovirus evolution and a better understanding of virus–host interactions that affect host morbidity and epidemiology.

## Figures and Tables

**Figure 1 microorganisms-11-02885-f001:**
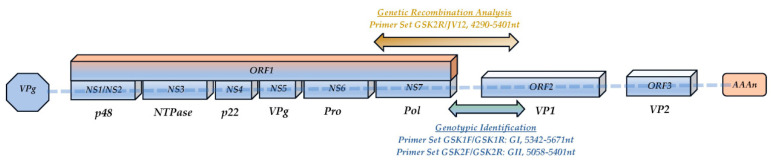
Primers used for norovirus genotyping and genetic recombination analysis. The relative position of the primers on norovirus reference strain “Norwalk” (GenBank Accession No. M87661) is also indicated.

**Figure 2 microorganisms-11-02885-f002:**
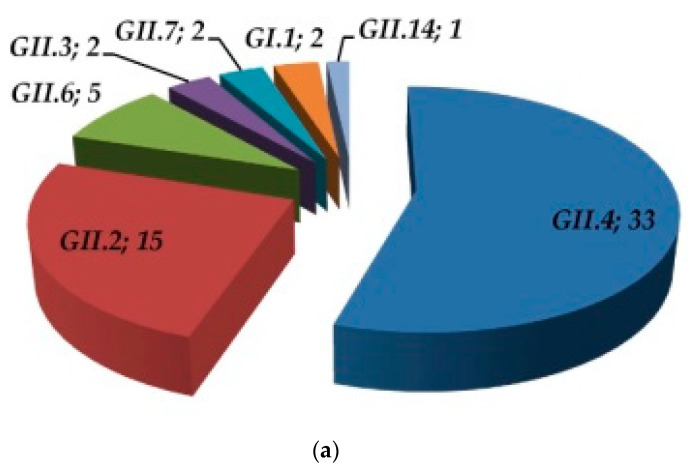
(**a**) Number of strains classified to each of the different norovirus genotypes identified and (**b**) the chronological pattern of genotype detection throughout the study period.

**Figure 3 microorganisms-11-02885-f003:**
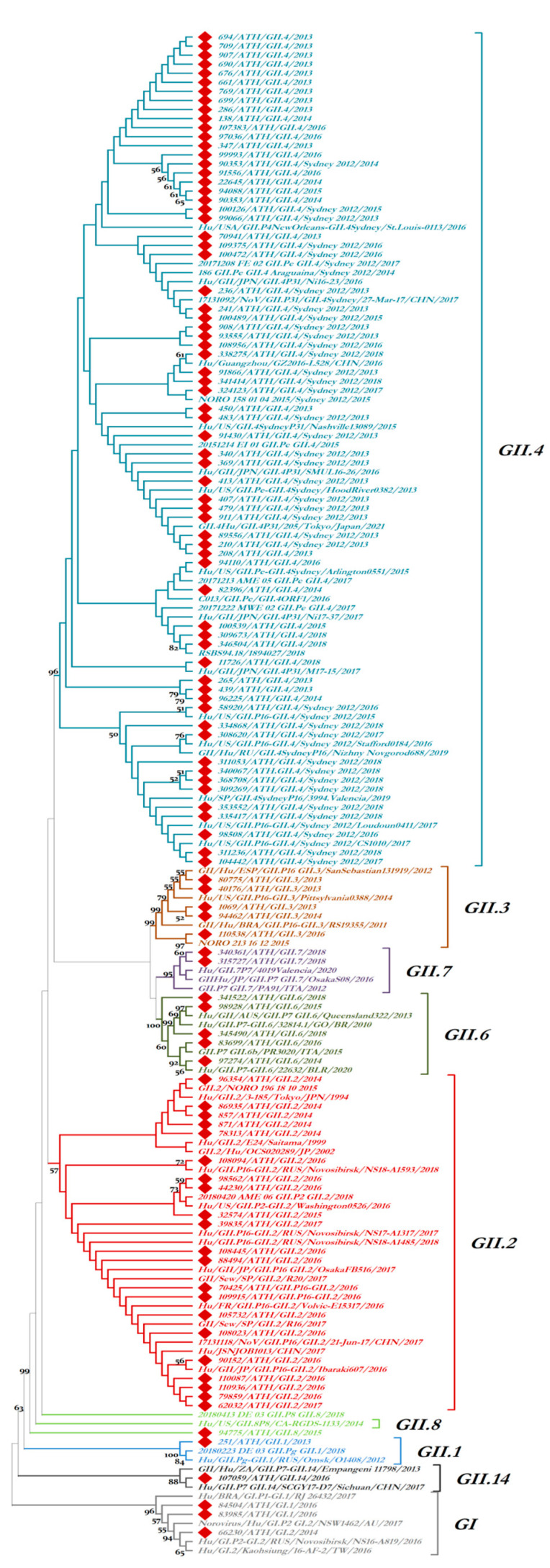
Dendrogram showing the phylogenetic comparison of strains detected in the present study with closely related norovirus strains of the same genotype that circulated either in Greece or worldwide on the basis of their partial ORF2 sequences. Bootstrap values are shown as percentages. Strains detected in Greece during the present study or in the past are marked with a rhombus (♦).

**Figure 4 microorganisms-11-02885-f004:**
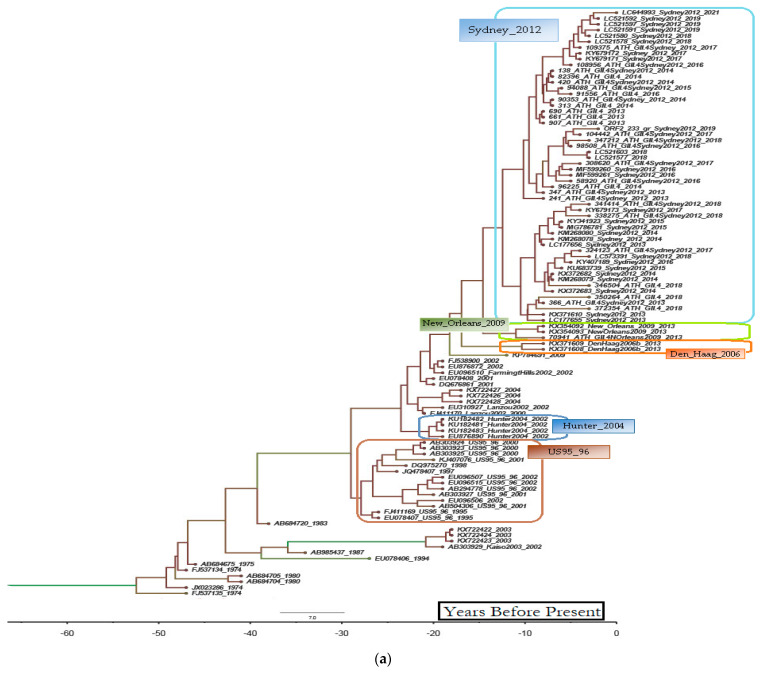
Temporal history of GII.4 (**a**) and GII.2 (**b**) noroviruses that have circulated globally during the last 50 years on the basis of partial ORF2 gene sequences. Strains identified during the present study are also included. The two GII.2 clusters that were observed to diverge approximately 20 years ago are indicated by a blue and a purple vertical line, respectively.

**Figure 5 microorganisms-11-02885-f005:**
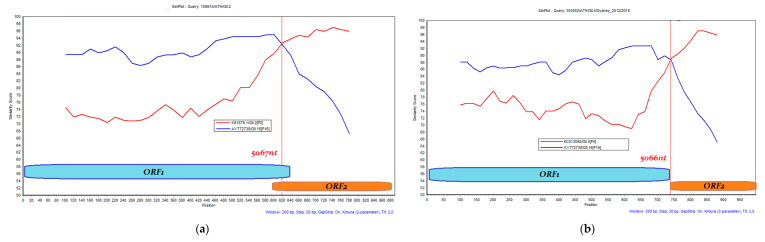
Representative similarity plots for each of the five recombination patterns that were observed during the present study. (**a**) GII.2[P16], (**b**) GII.4[P16], (**c**) GII.4[P21], (**d**) GII.4[P31], (**e**) GII.6[P7]. The percentage of nucleotide sequence identity is shown on the vertical axis. The predicted recombination site and its position on the norovirus genome are shown by the red vertical line.

**Table 1 microorganisms-11-02885-t001:** Norovirus genotypes of the strains that were detected with respect to both the partial ORF1 (RdRp) and ORF2 (VP1) nucleotide sequences. GenBank Accession Nos, as well as information about genetic recombination events and their breakpoints on the norovirus genome, are also included.

GenBank Accession No.	Position on NoV Genome	Isolate	Stool Collection Date	Genotype ORF2 (VP1)	Genotype ORF1 (RdRp)	Recombination Breakpoint Position (nt)	GII.4 Pandemic Variant
MT126392.1	5047–5309	83699/ATH/GII.6	Jan-2016	GII.6	Not Available	-	-
MT126388.1	5049–5349	109375/ATH/GII.4	Jan-2016	GII.4	Not Available	-	Sydney_2012
MT126407.1	5047–5344	107059/ATH/GII.14	Jun-2016	GII.14	Not Available	-	-
MT126405.1	5047–5343	58920/ATH/GII.4	Aug-2016	GII.4	Not Available	-	Sydney_2012
MT126384.2	5047–5326	90152/ATH/GII.2	Sep-2016	GII.2	Not Available	-	-
MT126402.1	5047–5339	108956/ATH/GII.4	Sep-2016	GII.4	Not Available	-	Sydney_2012
MT126389.1	5047–5337	98562/ATH/GII.2	Sep-2016	GII.2	Not Available	-	-
MT126387.1	4327–5351	109915/ATH/GII.2	Oct-2016	GII.2	GII.P16	5067 (MG746035 Ref.)	-
MT126393.1	5047–5353	108094/ATH/GII.2	Oct-2016	GII.2	Not Available	-	-
MT126398.1	5047–5344	100472/ATH/GII.4	Oct-2016	GII.4	Not Available	-	Sydney_2012
MT129796.1	5326–5627	84504/ATH/GI.1	Oct-2016	GI.1	Not Available	-	-
MT129795.1	5326–5609	83985/ATH/GI.1	Oct-2016	GI.1	Not Available	-	-
MT126395.1	5047–5344	105732/ATH/GII.2	Nov-2016	GII.2	Not Available	-	-
MT126397.2	4446–5349	94110/ATH/GII.4	Nov-2016	GII.4	GII.P21 (GII.Pb)	5029 (MH218671 Ref.)	Sydney_2012
MT126396.2	4326–5337	44230/ATH/GII.2	Nov-2016	GII.2	GII.P2	-	-
MT126408.1	4836–5356	110087/ATH/GII.2	Nov-2016	GII.2	GII.P16	5067 (MG746035 Ref.)	-
MT126401.1	4506–5328	107383/ATH/GII.4	Nov-2016	GII.4	GII.P31 (GII.Pe)	5019 (MK789435 Ref.)	Sydney_2012
MT126391.1	4323–5339	98508/ATH/GII.4	Nov-2016	GII.4	GII.P16	5066 (MK753032 Ref.)	Sydney_2012
MT126385.1	5047–5338	110538/ATH/GII.3	Nov-2016	GII.3	Not Available	-	-
MT126379.1	4384–5337	88494/ATH/GII.2	Nov-2016	GII.2	GII.P16	5067 (MG746035 Ref.)	-
MT126400.1	4337–5349	97036/ATH/GII.4	Dec-2016	GII.4	GII.P31 (GII.Pe)	5019 (MK789435 Ref.)	Sydney_2012
MT126399.1	5047–5349	99993/ATH/GII.4	Dec-2016	GII.4	Not Available	-	Sydney_2012
MT126404.1	5047–5349	91556/ATH/GII.4	Dec-2016	GII.4	Not Available	-	Sydney_2012
MT126390.1	5063–5352	103242/ATH/GII.6	Dec-2016	GII.6	Not Available	-	-
MT126394.2	5047–5344	110936/ATH/GII.2	Dec-2016	GII.2	Not Available	-	-
MT126378.1	4447–5349	108445/ATH/GII.2	Dec-2016	GII.2	GII.P16	5067 (MG746035 Ref.)	-
MT126377.1	5047–5337	79859/ATH/GII.2	Dec-2016	GII.2	Not Available	-	-
MT126406.1	5047–5322	108023/ATH/GII.2	Dec-2016	GII.2	Not Available	-	-
MT126386.1	4797–5312	70425/ATH/GII.2	Dec-2016	GII.2	GII.P16	5067 (MG746035 Ref.)	-
MT126382.1	5047–5329	104442/ATH/GII.4	Jan-2017	GII.4	Not Available	-	Sydney_2012
MT126381.1	5047–5337	39835/ATH/GII.2	Jan-2017	GII.2	Not Available	-	-
MT126380.1	5047–5328	62032/ATH/GII.2	Jan-2017	GII.2	Not Available	-	-
MT126383.1	5047–5349	308620/ATH/GII.4	Jun-2017	GII.4	Not Available	-	Sydney_2012
MT126403.1	5047–5328	324123/ATH/GII.4	Jul-2017	GII.4	Not Available	-	Sydney_2012
OP557585.1	5047–5325	11726/ATH/GII.4/2018	Feb-2018	GII.4	Not Available	-	Could not assign
OP557581.1	4425–5189	306792/ATH/GII.4/Sydney_2012/2018	Mar-2018	GII.4	GII.P16	5066 (MK753032 Ref.)	Sydney_2012
OP557589.1	5044–5349	309673/ATH/GII.4/2018	Apr-2018	GII.4	Not Available	-	Could not assign
OP557580.1	4333–5371	311236/ATH/GII.4/Sydney_2012/2018	Apr-2018	GII.4	GII.P16	5066 (MK753032 Ref.)	Sydney_2012
OP557587.1	4335–5348	309269/ATH/GII.4/Sydney_2012/2018	Apr-2018	GII.4	GII.P16	5066 (MK753032 Ref.)	Sydney_2012
OP557584.1	5048–5327	311053/ATH/GII.4/Sydney_2012/2018	May-2018	GII.4	Not Available	-	Sydney_2012
OP557579.1	5048–5299	315727/ATH/GII.7/2018	May-2018	GII.7	Not Available	-	-
OP557592.1	4346–5232	332648/ATH/GII.6/2018	Jun-2018	GII.6	GII.P7	5007 (*MW661284 Ref*.)	-
OP557591.1	5047–5356	335417/ATH/GII.4/Sydney_2012/2018	Jun-2018	GII.4	Not Available	-	Sydney_2012
OP557590.1	5047–5356	334868/ATH/GII.4/Sydney_2012/2018	Jun-2018	GII.4	Not Available	-	Sydney_2012
OP557572.1	4329–5362	338275/ATH/GII.4/Sydney_2012/2018	Jun-2018	GII.4	GII.P31 (GII.Pe)	5019 (MK789435 Ref.)	Sydney_2012
OP557593.1	4506–5331	341522/ATH/GII.6/2018	Jul-2018	GII.6	GII.P7	5007 (MW661284 Ref.)	-
OP557582.1	4376–5308	340067/ATH.GII.4/Sydney_2012/2018	Jul-2018	GII.4	GII.P16	5066 (MK753032 Ref.)	Sydney_2012
OP557575.1	4433–5363	341414/ATH/GII.4/Sydney_2012/2018	Jul-2018	GII.4	GII.P31 (GII.Pe)	5019 (MK789435 Ref.)	Sydney_2012
OP557574.1	5047–5313	340361/ATH/GII.7/2018	Jul-2018	GII.7	Not Available	-	-
OP557594.1	5066–5309	347211/ATH/GII.4/Sydney_2012/2018	Aug-2018	GII.4	Not Available	-	Sydney_2012
OP557586.1	4314–5366	353552/ATH/GII.4/Sydney_2012/2018	Aug-2018	GII.4	GII.P16	5066 (MK753032 Ref.)	Sydney_2012
OP557583.1	4353–5364	345490/ATH/GII.6/2018	Aug-2018	GII.6	GII.P7	5007 (MW661284 Ref.)	-
OP557571.1	4390–5363	346504/ATH/GII.4/2018	Aug-2018	GII.4	GII.P31 (GII.Pe)	5014 (KC175323 Ref.)	Could not assign
OP557570.1	4436–5298	347212/GII.4/Sydney_2012/2018	Aug-2018	GII.4	GII.P16	5066 (MK753032 Ref.)	Sydney_2012
OP557569.1	5048–5304	350264/ATH/GII.4/2018	Aug-2018	GII.4	Not Available	-	Could not assign
OP557578.1	5048–5307	372354/ATH/GII.4/2018	Sep-2018	GII.4	Not Available	-	Could not assign
OP557577.1	4323–5266	356299/ATH/GII.4/Sydney_2012/2018	Sep-2018	GII.4	GII.P16	5066 (MK753032 Ref.)	Sydney_2012
OP557576.1	5105–5326	372382/ATH/GII.3/2018	Sep-2018	GII.3	Not Available	-	-
OP557588.1	4326–5324	368708/ATH/GII.4/Sydney_2012/2018	Sep-2018	GII.4	GII.P16	5066 (MK753032 Ref.)	Sydney_2012
OP557573.1	4340–5221	358253/ATH/GII.4/Sydney_2012/2018	Sep-2018	GII.4	GII.P16	5066 (MK753032 Ref.)	Sydney_2012

## Data Availability

Data are contained within the article and Appendix A.

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
