# Peer review of "Predominance of Recombinant Norovirus Strains in Greece, 2016–2018"

_microorganisms, 2023, doi:10.3390/microorganisms11122885_

Round 1

Reviewer 1 Report

Comments and Suggestions for Authors

The current manuscript is of scientific interest since it describe a huge amount of data in order to provide a better understand regarding the molecular epidemiology and prevalence of norovirus in South Greece.  The manuscript provides a clear and concise presentation of the study's findings, particularly regarding the patterns of norovirus (NoV) circulation in Greece, and also covers a substantial time period (33 months), allowing for a comprehensive analysis of NoV circulation patterns and genotypic changes over time. Furthermore, the manuscript highlights novel findings related to NoV epidemiology in Greece, especially regarding the abrupt changes in NoV genotype predominance and the prevalence of recombinant strains. These findings contribute to a better understanding of NoV dynamics and molecular epidemiology.  The understanding of the prevalence and genetic diversity of NoV strains is crucial for public health efforts, especially in the context of gastroenteritis outbreaks. The manuscript underscores the importance of continuous genomic surveillance.

However, some comments are listed below in order to strengthen the manuscript content: 

1. Abstract: The abstract needs to be less descriptive in terms of results. Please write a brief introduction, objective, methods, results and the most important contribution of the manuscript as conclusion 

2. Ethical aspect: As human samples were collected and used in the study, it is important that the authors include a paragraph informing the use of the patient consent form allowing sample collection and use in the referred scientific work, as well as the protocol number of the research approved by the local ethical committee. 

2. Lack of Comparison: While the study presents interesting findings about NoV circulation in Greece, it would benefit from a more thorough discussion and comparison with other regional or global studies. This would help contextualize the results and highlight whether the observed patterns are unique to Greece or consistent with broader trends.

3. Data Interpretation: The manuscript discusses the presence of recombinant strains and their significance, but could provide more in-depth analysis or potential implications of these findings for NoV epidemiology and control.

4. Temporal analysis: Please provide a more detailed description for the timescale phylogeny by informing the used dataset, temporal signal evaluation by the root-to-tip analysis results (e.g. using TempEst program in Beast package), informing the parameters used for MCC tree construction, model selection and temporal tree construction using the Bayesian method. Furthermore, provide in a supplementary material;i) a table listing the used sequences, as well as a table with the likelihood logs related to the model of choice (in comparison to other models) to reconstruct the MCC tree and temporal analysis. 

Comments on the Quality of English Language

Minor english corretions are needed in order to improve the quality of the text. 

Author Response

Thank you very much for your significant comments on our manuscript. Our responses to your comments are the following:

Comment 1: The abstract was thoroughly changed, as suggested.

Comment 2: Neither informed consent, nor approval by an ethical committee were obtained before the study for three reasons: First of all, the study was carried out in retrospect, several years after collection of the samples and did not involve prospective evaluation. Secondly, all patient data were entirely anonymous and finally, clinical sample collection did not involve any invasive procedures since the samples constituted human excreta which were used for pathogen, i.e. non-human, genetic material detection and further characterization.

Comment 3: Comparison with a previous report on norovirus circulation in Greece is provided at the Discussion section, 1st paragraph, lines 269-271. Additional information about previous detection and characterization of recombinant norovirus strains in Greece was added at the Discussion section, page 6, 2nd paragraph (lines 293- 301).

Lines 223-229 and Figure 2 at the Results section report on the nucleotide sequence similarity and subsequent phylogenetic relationhip between the norovirus strains of the present study and norovirus strains that circulated worldwide in outbreaks and sporadic cases of gastroenteritis during the same time period, either in outbreaks and sporadic cases of gastroenteritis, or in the environment.

Lines 274-286 discuss how the lack of the seasonal pattern of norovirus detection in our study contradicts with other reports on the seasonal pattern of norovirus circulation, although those reports represent different geographic areas and vary with respect to the season of peak circulation.

Lines 309-326 compares the results about the prevalence of GII.2[P16] during a certain time period of the present study and global emergence of these recombinant noroviruses during approximately the same time period.

Lines 343-346 mention that recombinant GII.4[P16] strains, like those reported during the present study, were also recorded in USA in 2015 and that unclassified variants of GII.4 strains have been reported before either in Greece, or elsewhere in the world.

Comment 4: Following the reviewer’s recommendation, a last paragraph in the Discussion section was added which remarks on the fact that we still need a better understanding of the mechanism of norovirus emergence and persistence, in order to design and implement appropriate strategies for the control of norovirus infection.

Comment 5: One supplementary document is now provided that contains information about the isolates that were used, the year of their detection and their GenBank Accession Nos for GII.4 and GII.2 norovirus strains respectively. The temporal signal evaluation for each MCC phylogenetic tree using the root-to-tip analysis implemented in the TempEst software is also provided, as well as information about the the parameters used for MCC tree construction, model selection and temporal tree construction using the Bayesian method.

All changes made are highlighted in the manuscript, including additional references used. An improved version of the phylogenetic tree shown in Figure 4b was also added. We also revised the title of the manuscript, which now is: "Predominance of Norovirus recombinant strains in Greece, 2016-2028".

Reviewer 2 Report

Comments and Suggestions for Authors

The study from Siafakas and colleagues is an important addition in the surveillance of recombinant strains of norovirus. The findings are not completely novel as they are similar to those reported in previous studies, but nevertheless are reassuring in some key points as the circulation of the GII.2 geneotype.  

Specific comments:

1. The authors should consider omitting the abbreviation of norovirus "NoV" in the text as was recommended by the International Calicivirus Committee

2. The second and third paragraph of the Introduction could be improved or removed as the main point of the study is describe the process of recombination of norovirus.

3. The figure 2 (b) could be improved removing the labels inside the figure.  

Author Response

Thank you very much for your significant comments on our manuscript. Our responses to your comments are the following:

Comment 1: The abbreviation "NoV" of noroviruses was omitted throughout the text.

Comment 2: The objective of the study was not to describe, or discuss about the proposed models of recombination processes in noroviruses, but rather to observe the changing patterns of norovirus epidemiology, following from a previous report on the molecular epidemiology of noroviruses in South Greece a few years ago. And since recombination events are central to norovirus epidemiology, the present study extended its focus on the search of recombinant norovirus strains as well.

For this reason, the introduction is structured as follows:

1st paragraph, general information about norovirus taxonomy and biology, with a focus on genomic structure and the significance of capsid – coding region in norovirus evolution.

2nd paragraph: The “secrets” of norovirus widespread circulation and evolutionary success.

3rd paragraph: Clinical and epidemiological significance of noroviruses.

4th paragraph: Mutation and genetic recombination as driving forces of norovirus evolution.

5th paragraph: Variability amongst different norovirus genotypes regarding the extent of genetic variability and its role in the emergence and prevalence of specific genotypes, along with changing patterns in genotype and variant prevalence during the last decades.

6th paragraph: Conclusion about the significance of continuous surveillance for the changing patterns of norovirus molecular epidemiology and the objectives of the present study.

Comment 3: The labels in Figure 2(b) were removed

All changes made are highlighted in the manuscript, including additional references used. An improved version of the phylogenetic tree shown in Figure 4b was also added. We also revised the title of the manuscript, which now is: "Predominance of Norovirus recombinant strains in Greece, 2016-2028".

Reviewer 3 Report

Comments and Suggestions for Authors

This is a well written report on the genotyping and recombination of norovirus in south greece

The manuscript has a very solid content. It also well described the limitations of the study.

I have several suggestions which may improve the quality of the manuscript.

1. Please provide more information regarding global burden of norovirus. It would better to include the recent reference.

- https://www.cdc.gov/norovirus/burden.html

2. If possible, please describe a previous sporadic cases and outbreaks of norovirus in Greece in Introduction and Discussion section.

3. There are typo error(page 6, lane 11).

- It seems to figure 2 instead of figure 1.

Author Response

Thank you very much for your remarks about the manuscript. Our responses to your comments are the following:

Comment 1: Information about the global burden of norovirus infection is provided in the 3rd paragraph of the manuscript (lines 67-75).

Comment 2: Comparison with a previous report on norovirus circulation in Greece is provided at the Discussion section, 1st paragraph, lines 269-271. Additional information about previous detection of recombinant norovirus strains in Greece was added at the Discussion section, page 6, 2nd paragraph (lines 293- 301).

Comment 3: This was changed to Figure 2.

All changes made are highlighted in the manuscript, including additional references used. An improved version of the phylogenetic tree shown in Figure 4b was also added. We also revised the title of the manuscript, which now is: "Predominance of Norovirus recombinant strains in Greece, 2016-2028".